# Synthetic asters as elastic and radial skeletons

Qingqiao Xie[1], Xixi Chen[2], Tianli Wu[2], Tiankuo Wang[3], Yi Cao [3], Steve Granick[4,5], Yuchao Li[2]* & Lingxiang Jiang [1,4]*

The radial geometry with rays radiated from a common core occurs ubiquitously in nature for its symmetry and functions. Herein, we report a class of synthetic asters with well-defined core-ray geometry that can function as elastic and radial skeletons to harbor nano- and microparticles. We fabricate the asters in a single, facile, and high-yield step that can be readily scaled up; specifically, amphiphilic gemini molecules self-assemble in water into asters with an amorphous core and divergently growing, twisted crystalline ribbons. The asters can spontaneously position microparticles in the cores, along the radial ribbons, or by the outer rims depending on particle sizes and surface chemistry. Their mechanical properties are determined on single- and multiple-aster levels. We further maneuver the synthetic asters as building blocks to form higher-order structures in virtue of aster-aster adhesion induced by ribbon intertwining. We envision the astral structures to act as rudimentary spatial organizers in nanoscience for coordinated multicomponent systems, possibly leading to emergent, synergistic functions.

[1] College of Chemistry and Materials Science, Jinan University, Guangzhou 510632, China. [2] Institute of Nanophotonics, Jinan University, 511443 Guangzhou, China. [3] Collaborative Innovation Center of Advanced Microstructures, Department of Physics, National Laboratory of Solid State Microstructure, Nanjing University, Nanjing, China. [4] Center for Soft and Living Matter, Institute for Basic Science (IBS), Ulsan 44919, Republic of Korea. [5] Departments of Chemistry and Physics, UNIST, Ulsan 44919, Republic of Korea. *email: liyuchao@jnu.edu.cn; jianglx@jnu.edu.cn

Despite much success in fabrication of individual nanoparticles with almost any shape, material, and composition, limited progress has been made towards the organization and integration of multiple components in space and time to function in unison[1]. Looking for a positioning scheme, we notice the radial geometry with rays radiated from a common core that occurs ubiquitously in nature for its symmetry and functions, such as flowers for reproduction and hedgehog spines to defend themselves. In particular, intracellular microtubule asters act as radial skeletons to, among other functions, sustain deformation and keep organelles in desired place, underlying many cellular processes including signaling, polarization, and outgrowth (Fig. 1a)[2–7].

In the synthetic world, there have been a number of reports on microscopic core-ray particles since the early "morphosynthesis" time, such as mesoporous silica featuring radial patterns and inorganic/organic hybrid particles of neuron-like morphology[8,9]. More recently, the paradigms have been shifted to the emerging functionality of the radial geometry. For example, a micro-sculpture with a hemi-astral geometry functions as a beamsplitter to guide light from the core to rays;[10] TiO₂ spiky particles in contact with cells can physically activate innate immunity, echoing the effect of spiky nanostructures on virus or bacteria surfaces[11]. But the reported structures generally fall short to function as skeletons in terms of mechanical properties and positioning capability. For instance, radial particles of dendritic or spherulitic crystallinities with flower-like or hedgehog morphologies are too rigid to deform or too compact to accommodate other nanoparticles[12–20].

In this paper, we fabricate a class of synthetic asters with well-defined core-ray morphology that can function as elastic, radial skeletons to harbor nano- and microparticles (Fig. 1b–e). The asters are produced in a simple, robust, and high-yield step—the supramolecular self-assembly of a cationic gemini surfactant [ethylene-1,2-bis(cetyldimethylammonium)] with chiral counterions (a mixture of D- and L-tartrate) in water upon cooling

(Fig. 1c). Their formation proceeds with multiple stages, in which surfactant micelles first aggregate to form amorphous cores that gradually transform into divergently growing, twisted crystalline ribbons. The asters are of elasticity ~5 kPa and the ribbons are semiflexible filaments with a persistence length ~1 mm. The astral ribbons are packed at a moderate density to accommodate particles and to enable a simple positioning scheme by discriminating particle sizes and surface chemistry. Moreover, we discuss the similarities and differences between the current asters and the biological ones.

## Results

**Fabrication and morphology of synthetic asters.** We fully dissolve the surfactant powder[21,22] in water at 80 °C and then incubate the samples at 25 °C, during which precipitates appear to cloud the samples and then sediment (Fig. 2a and Supplementary Movie 1). The precipitates can be dispersed by gentle shaking to give a suspension surprisingly stable for days, whereas common precipitates this large would quickly sediment. Inspection by optical microscopy (OM) reveals prevailing asters featuring well-defined core-ray structure and large sizes ~50–120 μm (Fig. 2b). Each aster is a loose skeleton that occupies a spherical volume containing ~99.7% water. For asters in close contact, the semiflexible rays from two neighbors intertwin to form a visible "wall" in the middle plane preventing them from interpenetrating (Fig. 2c). We suspect that the freshly formed asters are somewhat interdigitating and densely packed in the precipitates and that the dispersed asters marginally land on each other with little interpenetration, thus sustaining their own weight against complete sedimentation. Since the asters can be robustly prepared in a wide range of surfactant concentrations and counterion ratios (Supplementary Figs. 1 and 2) in a single step with high yield (the precipitates are exclusively asters), we thus expect their production to be readily scaled up.

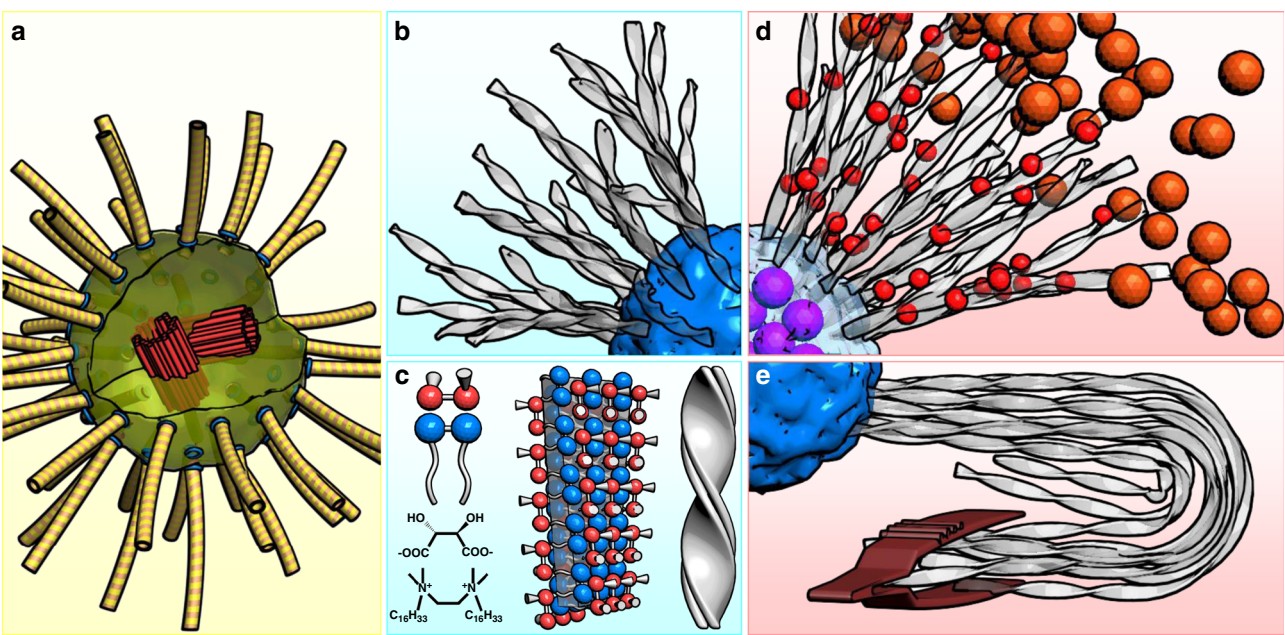

**Fig. 1** Schematic representations of microtubule and synthetic asters. **a** A pair of centrioles (red) assemble pericentriolar material to form a centrosome (light green) with γ-tubulin rings (blue) on its surface to initiate microtubule formation (yellow), producing a microtubule aster. **b**, **c** A gemini surfactant with a chiral counterion self-assemble into synthetic asters with an amorphous core and twisted ribbons made of crystalline bilayers. **d** The aster can radially position particles according to respective sizes and surface chemistry. **e** The aster is highly elastic and deformable, featuring semiflexible ribbons that can be bent 180°

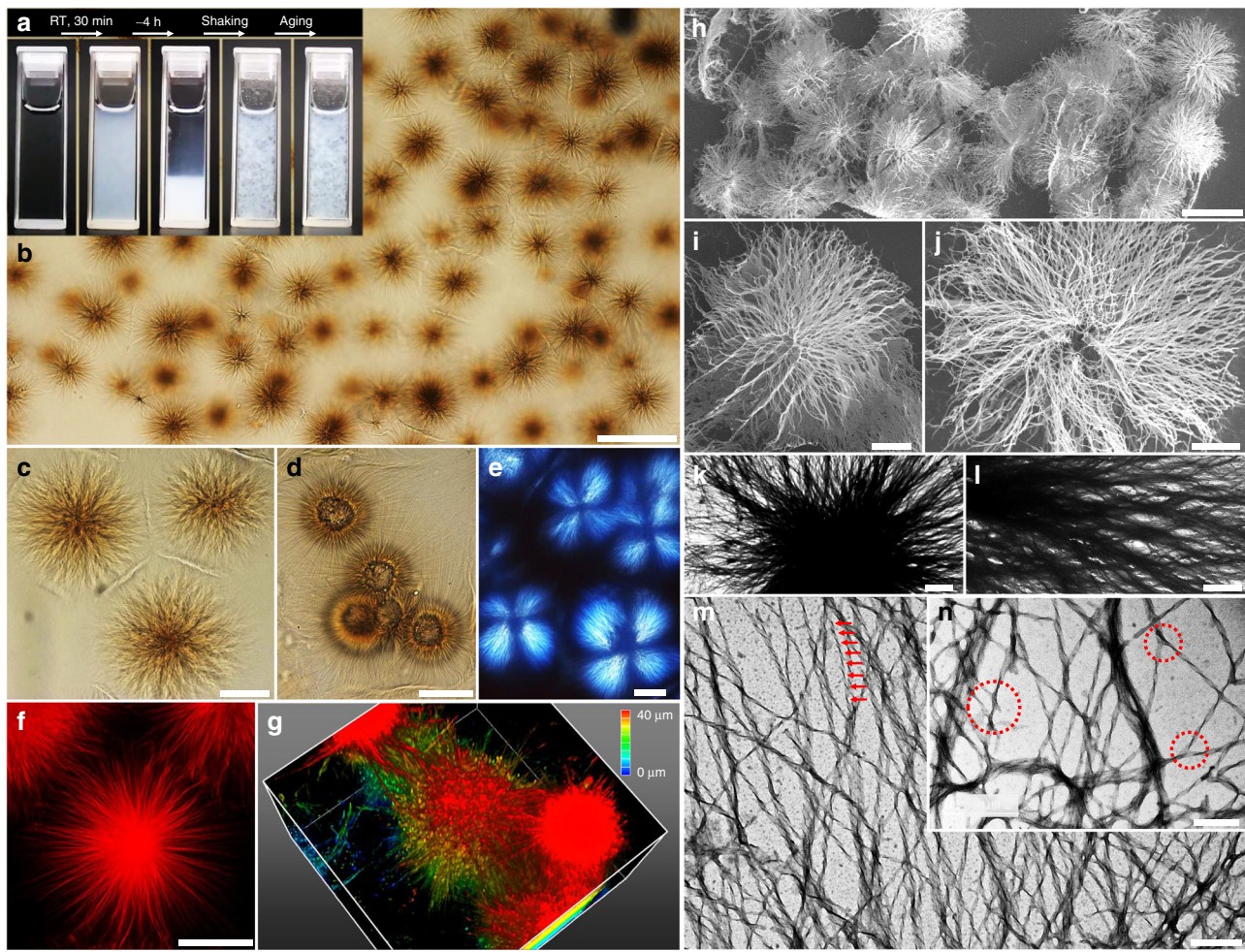

**Fig. 2** Core-ray morphology of the synthetic asters. **a** Pictures of asters formation in a cuvette; owing to their low volume-averaged density and non-interpenetrating nature, the gently dispersed asters resist complete sedimentation. **b–d** Images highlighting the prevalence of asters (**b**), boundaries between adjacent asters (**c**), and pronounced cores (**d**). **e** The asters appear as Maltese crosses with dark cores under cross polarizers. **f, g** Single asters (dyed by Nile red) are inspected by confocal microscopy in 2D (**f**) and 3D (**g**, box size = 60 × 60 × 40 μm³, color bar indicates height in Z-direction). **h–j** SEM pictures of the asters. **k–n** TEM pictures of the asters highlighting the core (**k**), inner rim (**l**), regular helicity in periphery (**m**), and branching junctions (**n**). Scale bars = 100 μm (**b**), 50 μm (**c–f**), 100 μm (**h**), 25 μm (**i, j**), 5 μm (**k**), 3 μm (**l**), 500 nm (**m**), and 300 nm (**n**)

At high concentrations, we observe asters with pronounced cores as large as 25 μm (Fig. 2d). Under a polarized optical microscope (POM), the asters manifest themselves as Maltase crosses with dark cores indicating anisotropic, crystalline rays and isotropic, amorphous cores (Fig. 2e). Fluorescently dyed, single asters are highlighted by confocal laser scanning microscopy (CLSM) in two dimension (2D) and three dimension (3D) (Fig. 2f, g). The structural details are further scrutinized by scanning and transmission electron microscopy (SEM and TEM, Fig. 2h–n); a dark core sits in the aster center (Fig. 2k) with ribbons radiating outwards (Fig. 2i, j, l). Notably, ribbons in the aster periphery can be recognized as twisted with a relatively monodisperse width (60 nm) and helicity in accord with work by Oda et al. (Fig. 2m)[21,22]. We also identify branched junctions (Fig. 2n). Gradual dissolution of the asters by heating is followed in Supplementary Fig. 3.

**Multi-stage mechanism of aster formation.** While it is not straightforward to understand how simple surfactant molecules (2 nm) assemble into complicated, large asters (~100 μm), the relatively long formation time (~3 h) allows us to follow the process ex situ and in situ, unveiling multiple intermediate stages with a distinct separation of length and time scales (Fig. 3). It is

known that many ionic surfactants have a Krafft point above which they stay in micelles and below which they tend to crystallize as their hydrophobic tails adopt an all-*trans* conformation[23]. For our gemini surfactant (Krafft point = 46 °C), cooling to 25 °C produces irregular aggregates with line widths a few times the micelle size (Fig. 3f), likely as a result of diffusion-limited aggregation of individual micelles (Fig. 1a, b). The contribution from surfactant monomers is negligible because merely ~0.1% of the surfactant molecules are in monomeric state in a typical preparation of 1 mg/ml surfactant (critical micelle concentration = 0.001 mg/ml). The micelle aggregates continue to grow into irregular particles (Fig. 3c, g, referred to as nodes) large enough to be observed by OM (Fig. 3j). We suspect the nodes to be solid, considering their nonspherical shape (Fig. 3g), and to be amorphous, considering no birefringence is observed under POM. We identify large spherical cores with dozens of nodes connected by randomly oriented ribbons (Fig. 3d, h, k), which presumably grow from the nodes, although we cannot rule out contribution from free micelles. These cores act as organizing centers in the final asters, where the crystalline ribbons grow in number and length with a clear radial orientation by consuming the amorphous nodes and the nodes disappear or retreat to a common center (Fig. 3e, i, l, m). The formation process is

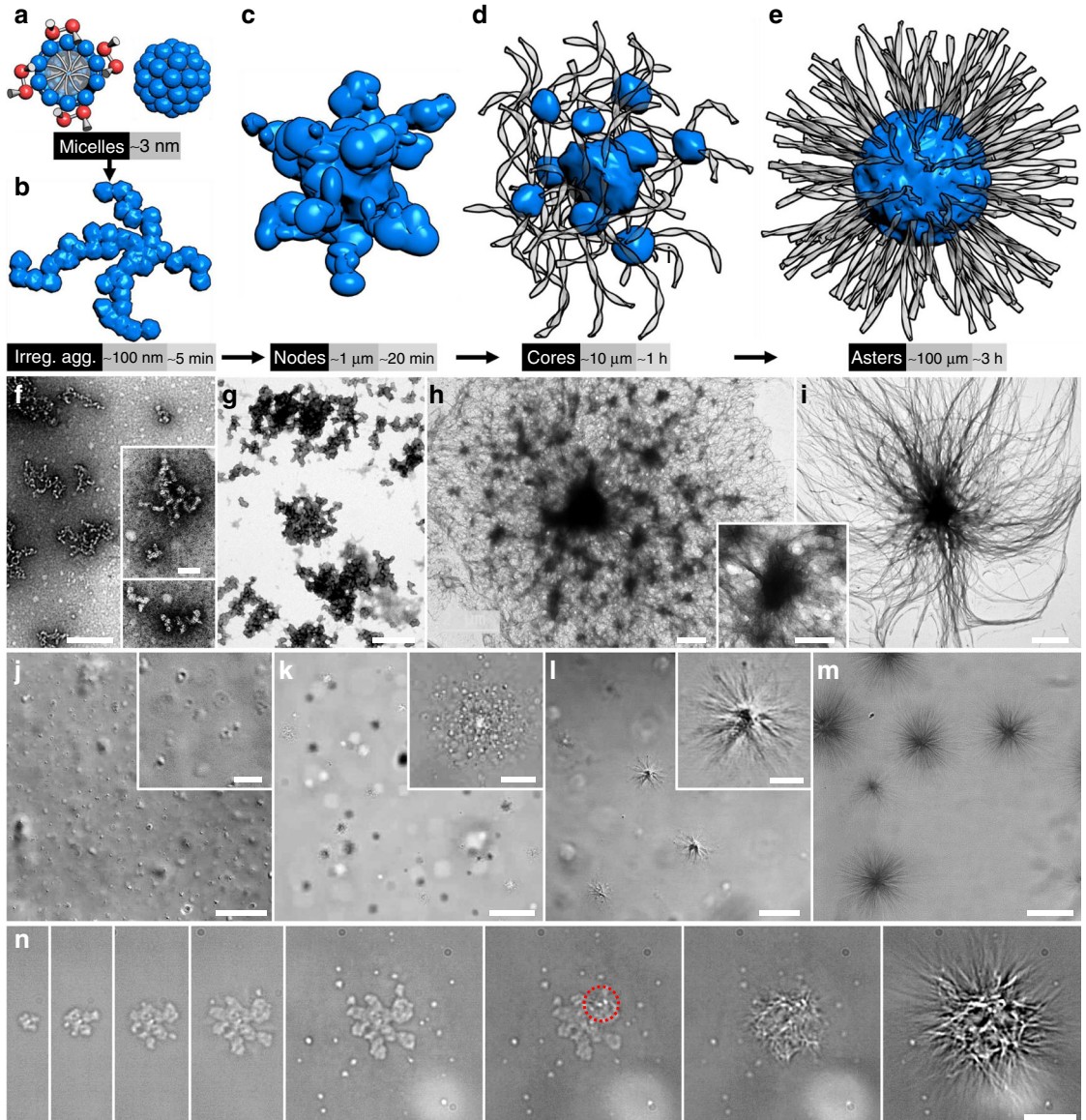

**Fig. 3** Formation kinetics of the asters. **a–e** Schematic illustrations of aster formation with a distinct separation of length and time scales. **f–i** Ex situ TEM observations of the irregular aggregates (**f**), nodes (**g**), cores (**h**), and asters (**i**). **j–m** In situ OM observations of the nodes (**j**), cores (**k**, interconnecting ribbons are unresolvable at this stage), asters (**l**, **m**). **n** Real-time formation of a single aster on glass surface, where fibrilization is initiated inside the red circle. Scale bars = 200 nm (**f**), 50 nm (**f**, inset), 500 nm (**g**), 1 μm (**h**), 500 nm (**h**, inset), 5 μm (**i**), 20 μm (**j**), 3 μm (**j**, inset), 30 μm (**k**), 3 μm (**k**, inset), 20 μm (**l**), 5 μm (**l**, inset), 50 μm (**m**), and 10 μm (**n**)

recorded in real time for a single aster and a fixed view in Supplementary Movies 2 and 3, respectively. Formation of a semi-2D aster on glass surface is slightly different; this case features an extensively growing, irregular core and multiple radiating centers in the final form (Fig. 3n and Supplementary Movie 4).

For synthetic molecules, their assembly into fibers is usually dictated by the classic nucleation-growth mechanism that, in the absence of organizing centers, would produce a percolating network (gels) of randomly arranged fibers with few exceptions[24,25]. In contrast to the classic mechanism, fiberization here is preceded by the relatively long-lived, yet metastable amorphous nodes (Fig. 3c). They play a crucial role in aster formation by constituting the organizing centers (cores, Fig. 3d) and eventually transforming into crystalline, radial ribbons. Although the core size varies, we do not observe any core-free asters. In addition, externally added hydrophobic particles can serve as heterogeneous nucleation sites facilitating core formation

around them. Adding too many particles (particle/surfactant weight ratio >0.1) produces small, incomplete asters. Therefore, the presence of cores at a proper concentration is crucial for aster formation in this system.

**Positioning capability.** We seek to establish a rudimentary scheme of positioning by virtue of nonspecific interactions between microparticles and the synthetic asters. Particles of different sizes and surface chemistry are mixed with the surfactant solution at 80 °C, followed by incubation at 25 °C that allows colocalization during aster formation. According to partitioning of the particles at different radial positions, they are categorized into four types (Fig. 4a, b). Type A are the hydrophobic particles that participate in aster nucleation (Fig. 4c–e), thus staying in the cores, while all the hydrophilic particles (type B to D) are expelled from the cores. The asters are significantly smaller and ribbons

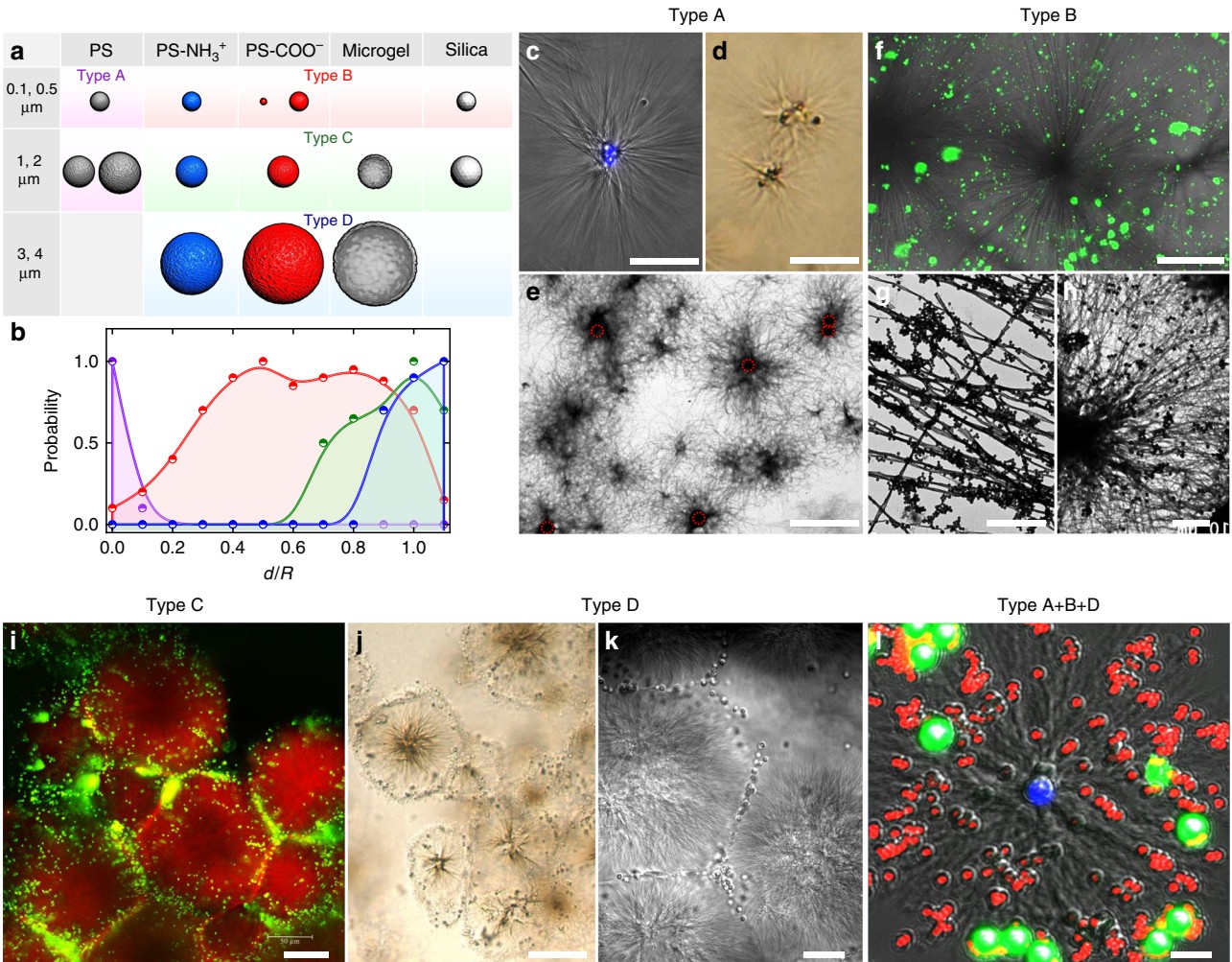

**Fig. 4** Positioning capability of the asters. **a** Categorization of the polystyrene (PS), amine-modified PS (PS-NH$_3^+$), carboxylate-modified PS (PS-COO$^-$), poly($N$-isopropylacrylamide) hydrogel (microgel), and silica particles into four types according to particle size and surface chemistry. The spherical representations are not scaled to size. **b** Statistics of spatial distributions of the particles along aster radius. Purple to blue denotes type A to D. **c–e** Type-A particles (2 μm PS in **c** and 1 μm PS in **d**, **e**) resides in the aster cores. **f–h** Type-B particles (0.1 μm PS-NH$_3^+$ in **f**, **g** and 0.5 μm PS-COO$^-$ in **h**) spread along the aster ribbons. **i** Type-C particles (1 μm PS-COO$^-$) are distributed near the outer rims. **j**, **k** Type-D particles (4 μm PS-COO$^-$ in **j** and 4 μm microgel in **k**) are exclusively localized along the peripheries. **l** Different particles are sorted into respective localizations (2 μm blue PS in the core, 0.5 μm red PS-COO$^-$ along the ribbons, and 4 μm green PS-COO$^-$ by the periphery). Scale bars = 50 μm (**c**, **d**), 5 μm (**e**), 50 μm (**f**), 2 μm (**g**), 5 μm (**h**), 50 μm (**i**), 100 μm (**j**), 20 μm (**k**), and 5 μm (**l**)

sparser in the presence of hydrophobic particles, possibly because they act as extra, heterogeneous nucleation centers. For the hydrophilic particles, type B are the small ones that distribute along the aster ribbons, where the particle–ribbon affinity is always high regardless of particle charge or surface chemistry (Fig. 4f–h). Type C are the mid-size particles that can barely be accommodated near the inner rims, but prefer to stay near the outer rims (Fig. 4i). Type D are the particles that are too large to fit in between the ribbons and are exclusively localized along the periphery (Fig. 4j, k).

Next, we test if the asters can simultaneously discriminate different particles and place them at their respective positions. In the presence of mixed particles, the asters localize type-A particles (blue) into the core, type-B particles (red) along the ribbons, and type-D particles (green) by the outer rims with no interference from each other (Fig. 4l). To quantitatively resolve particle distribution, we employ a particle tracking program (TrackMate[26] on ImageJ) to obtain particle coordinates from the fluorescence channel and identify aster center and radius $R$ from the transmission channel (Supplementary Fig. 4). Lateral distance

of a particle from the center $d$ is determined and statistics of $d/R$ for different types of particles is established (Fig. 4b). Clearly, from type A to D, the particles are sharply localized in the core, widely distributed along the ribbons, preferentially placed near the outer rims, and constrained to the periphery. This positioning scheme thus offers a facile means to compartmentalize mixed components, which are difficult to separate otherwise, in an efficient and orthogonal manner.

**Aster elasticity**. We conduct atomic force microscopy (AFM) indentation measurements on single asters, in which the common sharp tip is replaced by a 7-μm silica sphere to ensure smooth compression on loose astral ribbons (Fig. 5a, inset). Two typical indentation–retraction curves reveal Young's modulus $E = 6 \pm 4$ kPa (Fig. 5a). The finite hysteresis between black and gray data reflects an energy dissipation mainly caused by attachment of astral ribbons to the AFM tip during indentation and detachment during retraction. Distributions of $E$ can be approximated by Gaussian functions at each spot (curves in Fig. 5b). Notably, the average $E$,

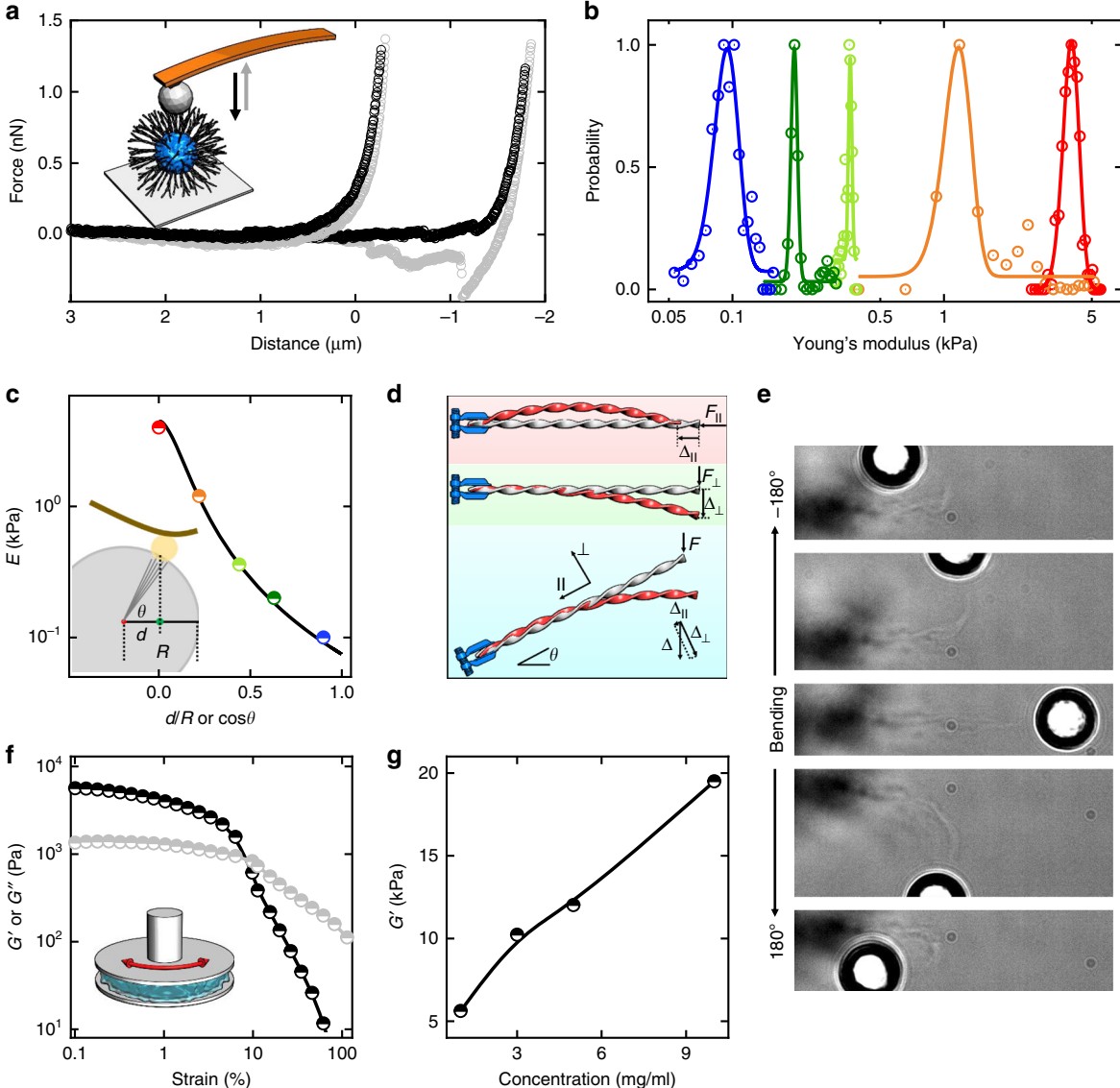

**Fig. 5** Mechanical properties of the asters. **a** A 7-μm silica sphere is attached to an AFM tip to probe the single asters; the indentation curves (black) can be fitted by the Hertz model, giving $E$ (Young's modulus) ~6 kPa. Small hysteresis of the retraction curves (gray) suggests a finite energy dissipation. Sometimes we observe negative forces during retraction (the right curve), reflecting adhesion of the tip sphere to the aster. The two curves are horizontally shifted for clarity. **b** $E$ drops significantly when the probed location is moved from the center to periphery (from blue to red). **c** Dependence of $E$ as a function of $d/R$. Points are experimental data and black curve is the fitting by the analytical function in the main text. The inset displays geometry of AFM measurements. **d** Schematic representation of ribbon deformation under force. **e** An aster sector is reversibly bent by an optical tweezer. The silica sphere is 10 μm in diameter. **f** Shear moduli ($G'$ and $G''$, black and gray points, respectively) of the aster precipitate phase (1 mg/ml) measured by a rheometer. **g** Variation of $G'$ as a function of surfactant concentration

probed at sub-aster locations, decrease markedly and regularly from the aster center to periphery (red to blue in Fig. 5b, c).

To rationalize this dependence, we put forward a simple model that is briefly sketched below (see Methods for details). Consider a single astral ribbon as a semiflexible filament of contour length $\ell$ and persistence length $\ell_p$ (Fig. 5d). In the linear regime, the filament is characterized by two spring constants transverse ($k_\perp$) and parallel ($k_\parallel$) to the filament with $k_\parallel \gg k_\perp$ [27,28]. The filament is clamped at one end with a fixed position and a fixed orientation $\theta$. A force $F$ in the gravity direction acting on the other end causes a deformation $\Delta$ in the force direction, and they are related by

$$\frac{F}{\Delta} = k_\parallel/(1 + \frac{2\ell_p}{\ell}\cos^2\theta). \qquad (1)$$

Clearly, the elastic response is a function of orientation $\theta$ for given $\ell$ and $\ell_p$. In an AFM measurement on an aster of radius $R$ with the probing spot at a lateral distance $d$ away from the astral center, the average orientation of ribbons in the probing spot is given by $\cos\theta = d/R$ (Fig. 5c, inset). Finally, the elastic modulus is estimated by an analytic form

$$E \approx A/[R^4(1 + \frac{2\ell_p}{R}\cos^2\theta)]. \qquad (2)$$

The prefactor $A$ represents parameters that are invariant to $\cos\theta$ like $\ell_p$ and $k_\parallel$, and factors that we do not explicitly consider but assume to be insensitive to $\cos\theta$, such as effective number of ribbons in contact with the cantilever tip. Given $R = 50\,\mu m$, a simple fitting matches the experimental data satisfactorily (black curve in Fig. 5c), leading to $\ell_p \approx 1\,mm$.

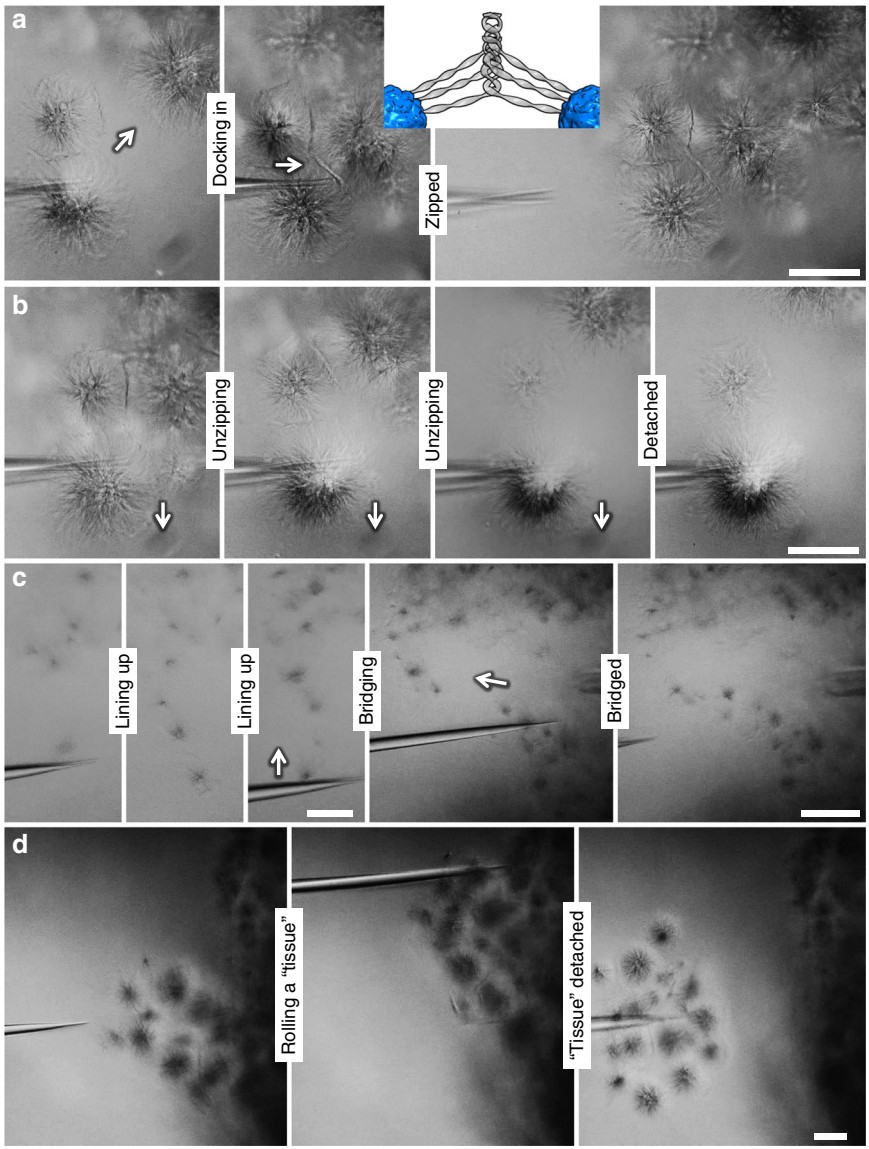

**Fig. 6** Micromanipulation of aster clusters. **a** When two asters are docked into a third one, the ribbons zip up in the boundaries (inset scheme) to form strong junctions. **b** The two asters are dragged away from the third one, causing unzipping and eventual detachment. **c** Asters are maneuvered to form a line and a bridge. **d** An aster cluster is rolled on a surface and then removed from the surface. Scale bars = 50 μm (**a–d**)

To further investigate the aster elasticity and deformability, we trap a 10 μm PS microsphere with an optical tweezer and use it as a handle to deform or move the asters. For example, an aster sector can be bent up to ± 180° by an attached microsphere (Fig. 5e and Supplementary Movie 5). Notably, the bending is reversible suggesting that the ribbons remain intact during the manipulation. The deformed aster sector can quickly restore its equilibrium shape within 1 s once the optical tweezer is removed (Supplementary Movie 6). Asters can also be maneuvered to move in solution in a controllable fashion (Supplementary Movie 7). The precipitate phase full of asters (initial concentration = 1 mg/ml) behaves like a typical hydrogel with $G'$ (storage modulus, ~5 kPa) dominating over $G''$ (loss modulus) up to 10% strain as measured by a rheometer (Fig. 5f). $G'$ increases from 5 to 20 kPa with higher initial concentrations (Fig. 5g).

**From single asters to higher-order structures**. Finally, we demonstrate how to micromanipulate these asters as building blocks to construct higher-order scaffolds on demand. In doing so, "zipper" interaction between asters plays a crucial role. We

dock two asters, using a silica fiber tip, onto a third one and then stress them until clear dark boundaries appear between the neighbors (Fig. 6a and Supplementary Movie 8). The semiflexible ribbons from the contacting asters are expected to intertwine like zippers, forming a strong aster–aster junction. The two asters are dragged continually, such that the junctions gradually unzip and the asters finally detach (Fig. 6b and Supplementary Movie 9). The zipping–unzipping cycles can be repeated several times until the peripheral ribbons are completely worn out. Although a few ribbons might be damaged during manipulation, the asters usually retain remarkable integrity. A line of five asters is formed by transferring asters to the line tip and this line is then connected to a lower cluster to bridge two peninsulas (Fig. 6c and Supplementary Movie 10). A roughly spherical cluster of asters is maneuvered to roll on a surface and then detach from the surface (Fig. 6d and Supplementary Movie 11).

**Discussion**

We note that the current synthetic asters are morphologically similar to microtubule asters in animal cells—formed by a

centrosome and radially arranged microtubules (Fig. 1a)[2–7]. Formation of the latter usually starts from a pair of centrioles that recruits an amorphous assembly of proteins (pericentriolar material) to form a microtubule-organizing center (named centrosome). The centrosome triggers nucleation of microtubules on its roughly spherical surface followed by radial growth (Fig. 1a)[29–31]. The minimal reconstitution of centrosomes in vitro recapitulated them as spherical, disordered[29] (or ordered)[30] condensates that can efficiently concentrate tubulins and trigger microtubule nucleation, where common physicochemical mechanisms beyond specific protein structures are at play[31]. In a few cases, asters have also been observed to form in the absence of centrosomes as a result of motor–microtubule interactions[32]. While the biological asters are constructed by multiple proteins as a group effort, the synthetic asters in this system are formed, remarkably, by a single component in a simple cooling step.

Among many functions, the biological asters serve as elastic skeletons to provide mechanical strength and as positioning scaffolds. Subcellular localizations of distinct organelles are realized by motor/anchoring proteins on the microtubules to transport/immobilize organelles based on specific interactions; for example, the Golgi apparatus is usually in proximity to the centrosome, the endoplasmic reticulum spreads throughout the cytoplasm, and certain signaling proteins are transported to the aster periphery[2,4]. In our case, the persistence length of astral ribbons is ~1 mm, close to that of microtubules ~4 mm; the elastic modulus of a single synthetic asters is ~5 kPa, comparable to that of a cell. Particle positioning by the synthetic asters is passive in nature, relying on the discrimination by particle size and surface chemistry.

Although we limit ourselves to the morphological, mechanical, and positioning aspects in this paper, the biological asters possess many features that are not addressed here. For example, the microtubules are polar, dynamic filaments that constantly undergo assembly at one end and disassembly at the other end by consuming ATP, whereas the current astral structures are of no polarity and are static in nature. The microtubules can serve as tracks of motor proteins for active transportation, while the current asters can hardly support active particles in the absence of specific particle–ribbon interactions. Similarities and differences between the biological and synthetic asters and a number of representative core-ray particles reported previously are summarized in Supplementary Table 1. The core-ray particles can be roughly categorized into inorganic, organic, and hybrid ones that are markedly different in size, ray density, elasticity, and functionality. The main advances of the current asters against previous particles are the mechanical properties (elasticity and deformability) and positioning capability.

We emphasize that the connections between the synthetic and biological asters are superficial and phenomenological, and that the former is, by no means, a complete mimicry of the latter. However, we do hope that the current asters can act as a rudimentary, prototype system into which more biomimetic functions can be incorporated.

The radially positioning scheme is envisioned to open new possibilities in nanoscience to establish spatially organized multicomponent systems; designed compartmentalization is recognized to have potential applications in catalysis, energy and mass transportation, and signaling, but how to implement functional compartments have been a stumbling block. Beyond the localization of particles by nonspecific interactions demonstrated here, we expect positioning of molecules or organelles to be possible based on their interactions with the amorphous versus crystalline surfactant, or alternatively by anchoring them to the colocalized particles with biotin–avidin or antibody–antigen linkages[33]. In terms of the aster architecture, this may be generalized using the rich variety of other fiber-forming molecules[34–36] based on the present insight on the organizing centers, which can be homogeneously formed or heterogeneously introduced. Moreover, the zipper interaction between asters shows a path towards assembly of precise, higher-order scaffolds of complicated geometries beyond the present radial one[37].

## Methods

**Materials**. 1-Bromohexadecane (98%) was purchased from Adamas. Silver carbonate (Ag$_2$CO$_3$, 99.9%), L-tartaric acid (99%), D-tartaric acid (99%), acrylic acid (>99.7%), N,N′-methylenebis(acrylamide) (≥99.0%), potassium persulfate (99.99%), and sodium chloride (≥99.5%) were purchased from Aladdin Chemicals. N,N,N′,N′-tetramethyl ethylenediamine (≥99.0%), N-isopropylacrylamide (98%), silica particles, and polystyrene (PS) particles were purchased from Macklin. Surface-modified PS particles were purchased from Thermo Fisher Scientific. Deionized water (18.2 MΩ·cm) was produced by a Milli-Q water purification system (Millipore, USA).

**Surfactant synthesis**. The surfactant was synthesized as reported[22]. N,N,N′,N′-tetramethylethylenediamine (0.1 mol) and 1-bromohexadecane (0.8 equiv.) were heated in MeCN (200 ml) at 40 °C for 1 day. Subsequent evaporation and crystallization from ethanol produce the gemini surfactant [ethylene-1,2-bis(cetyldimethylammonium)] with bromide counterions. The counterion exchange from bromide to tartate was performed at strict stoichiometry by mixing the surfactant with a suspension of silver salt of tartaric acid in methanol. Stoichiometry and purity of the product were confirmed by element analysis and nuclear magnetic resonance.

**General methods**. We weighed desired amount of surfactant powder in water (typically, surfactant concentration = 1 mg/ml and L-/D- counterion ratio = 2/1), dissolved the powder at 80 °C, and then incubated the sample at 25 °C for at least 4 h for the asters to be fully developed. In positioning experiments, microparticles were mixed with the surfactant solution at 80 °C and colocalization took place during aster formation at 25 °C. The asters were inspected by POM (Olympus BX51, ×40 objective), CLSM (Leica TCS SP8, ×60 oil objective, fluorescently dyed by Nile red), SEM (Zeiss EVO MA15, samples freeze dried), and TEM (Tecnai Spirit, negatively stained by uranyl acetate). Shear moduli of the aster precipitate phase were measured by a rotational rheometer (TA DHR-2 with a flat plate clamp at 10 rad/s). Throughout this work, temperature was kept at 25 °C, unless specified otherwise.

**Atomic force microscopy**. The experiments were carried out on an AFM (JPK Nanowizard II, JPK Instruments, Berlin, Germany) in water at room temperature. The spring constant of the cantilevers was in the range of 0.03–0.07 N/m. A 7-μm silica ball was attached onto the silicon nitride tip (type MLCT, Bruker Company) via the epoxy glue. A cantilever was positioned on top of an aster center, followed by approaching at a constant speed of 2 μm/s. The cantilever was then retracted and moved to another site to repeat the measurements. Young's modulus, E, is obtained by fitting the approaching curves by the Hertz model[38,39].

**Optical tweezer**. Optical tweezers are capable of trapping and manipulating single microspheres by exerting small optical forces (on the order of pico-newton) with a highly focused laser beam. Our optical tweezers system (Tweez250si) is based on an inverted Nikon microscope with a water-immersed, high numerical aperture objective (NA = 1.0 to focus the laser beam tightly for 3D trapping) and a 1064-nm laser with an optical power of 32 mW. An optical tweezer directly trapped an externally introduced microsphere that acted as a handle to manipulate the adhered asters.

**Manipulation by silica fiber tip**. The fiber tip was fabricated from a commercial multimode optical fiber by a flame-heating technique. First, the polymer jacket of the fiber was stripped off to obtain a bare fiber of 3.0 cm in length. The bare fiber was heated by the outer flame of an alcohol lamp for 50 s until the fiber reached its melting point. Then, the fiber was drawn at a rate of ~4 mm/s, which gradually tapers off. Finally, the diameter of the fiber tip shrank from 125 to 1.3 μm within an axial distance of 52 μm. This fiber tip was mounted on a tunable fiber micromanipulator (Kohzu Precision Co., Ltd.), allowing us to operate it in 3D at a precision of 50 nm.

**Model for astral elasticity**. First consider a single astral ribbon as a semiflexible filament of contour length $\ell$ and persistence length $\ell_p$ (Fig. 5d). In the linear regime, the filament is characterized by two spring constants transverse and parallel to the filament, respectively,

$$k_\perp = 3\kappa/\ell^3, \qquad (3)$$

$$k_\parallel = 6\kappa^2/(k_B T \ell^4), \tag{4}$$

where $\kappa$ is related to persistence length $\ell_p$ by

$$\ell_p = \kappa/(k_B T). \tag{5}$$

The filament is clamped at one end with a fixed position and a fixed orientation. A force acting on the other end can deform the filament. When the force is parallel and transverse to the filament, respectively, the deformation is given by

$$\Delta_\parallel = F_\parallel/k_\parallel, \tag{6}$$

$$\Delta_\perp = F_\perp/k_\perp. \tag{7}$$

When the filament is placed at an angle of $\theta$, $F$ is decomposed into $F_\parallel = F \sin\theta$ and $F_\perp = F \cos\theta$, and deformation in the force direction reads,

$$\Delta = \Delta_\parallel \sin\theta + \Delta_\perp \cos\theta = \frac{F \sin^2\theta}{k_\parallel} + \frac{F \cos^2\theta}{k_\perp}. \tag{8}$$

Combination of Eqs. 3–5 and 7 leads to

$$\frac{F}{\Delta} = k_\parallel/[1 + (\frac{2\ell_p}{\ell} - 1)\cos^2\theta]. \tag{9}$$

In our case, the filament is quite rigid so $\ell_p \gg \ell$ and Eq. 9 reduces to

$$\frac{F}{\Delta} = k_\parallel/(1 + \frac{2\ell_p}{\ell}\cos^2\theta). \tag{10}$$

Next, we consider the geometry of AFM measurements on a single aster (radius $R$) with the probing spot at a lateral distance $d$ away from the center. The average orientation of ribbons in the probing spot is given by $\cos\theta = d/R$. The observed elastic modulus ($E$) measures the ratio of stress to deformation,

$$E \propto \frac{F}{\Delta}. \tag{11}$$

Given Eq. 4 and $\ell = R$, we finally reach

$$E \approx A/[R^4(1 + \frac{2\ell_p}{R}\cos^2\theta)]. \tag{12}$$

The prefactor $A$ represents parameters that are invariant to $\cos\theta$ like $\ell_p$ and $k_\parallel$, and factors that we do not explicitly consider but assume to be insensitive to $\cos\theta$, such as effective number of ribbons in contact with the cantilever tip. Now we compare Eq. 12 (black curve in Fig. 5c) with the experimental results (color dots). In two limiting cases of $d \to 0$ and $d \to R$, the ratio of $E$ reads,

$$\frac{E_{d\to 0}}{E_{d\to R}} = \frac{2\ell_p + R}{R}. \tag{13}$$

For this particular aster, $R \approx 50\,\mu m$ and the ratio is 40 from Fig. 5c, giving $\ell_p \approx 1\,mm$. Notably, this persistence length is quite close to that of microtubules (4 mm). With $R$ and $\ell_p$ determined, the shape of the curve is fixed and $A$ merely shifts it vertically (Fig. 5c). A simple fitting matches the experimental points remarkably well.

## Data availability
The authors declare that all data supporting the findings of this study are available within the paper and its supplementary information files or from the corresponding authors upon reasonable request.

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

## Acknowledgements
L.J. acknowledges support by National Natural Science Foundation of China (No. 21773092), Guangdong Natural Science Funds for Distinguished Young Scholar (No. 2018B030306011), and the Fundamental Research Funds for the Central Universities (No. 21617320). S.G. acknowledges support by the Institute for Basic Science, project code IBS-R020-D1. Y.L. acknowledges support by National Natural Science Foundation

of China (No. 61905092) and the Fundamental Research Funds for the Central Universities (No. 11619321).

## Author contributions

Q.X., Y.L., and L.J. conceived the experiments. Q.X. synthesized the surfactant, discovered the asters, and carried out the general experiments. X.C. and T. Wu performed the micromanipulation experiments. T. Wang conducted the AFM experiments. Q.X., Y.L., Y.C., S.G., and L.J. analyzed the data. All the authors contributed to discussing the results and writing the paper.

## Competing interests

The authors declare no competing interests.
