## [Peer Review File · Nature Communications]

Reviewers' comments:

Reviewer #1 (Remarks to the Author):

The work presents a procedure for fabricating microscopic synthetic asters. The paper is technically sound, and the results are novel to justify publication in Nature Communications. However, a few questions need to be addressed prior to acceptance.

1. In addition to mechanical rigidity and positioning, biological asters have other important functions, such as polarity and active transport. Presumably, the synthetic asters also exhibit some sort of polarity, although this property is not explicitly characterized. However, in addition to the positioning of passive particles, it would be interesting to make a connection to the transport of active particles.
2. Throughout the paper is claimed that centrosomes are crucial for the formation of biological asters. While it is true, there are also experimental demonstrations of asters formed spontaneously even in the absence of centrosomes, Ref. 6. By the way, the name of the first author of Ref 6 is misspelled, it should be Nedelec. I am wondering if there is a similar pathway for synthetic asters.
3. The part on the mechanical properties of asters is rather incomplete. It is not clear what insights provides macroscopic rheology of aster precipitate phase since the elastic moduli depend on the aster concentration, positions, etc. The origin of hysteresis in Fig 4n is not clarified: I am not sure what does "a finite energy dissipation" mean in this context. Fig, 4o demonstrates a variation of the elastic modulus E as a function of position with respect to the center. Can a simple mechanical model explain this dependence?

Reviewer #2 (Remarks to the Author):

NCOMMS-19-18860

Synthetic asters as elastic, radial skeletons

This is a relatively short and straightforward paper that describes the experimental formation of aster-like supramolecular aggregates of a known Gemini surfactant in the presence of chiral counterions. The factors giving rise to the helical/twisted nature of the individual supramolecular ribbons have been extensively described (cited Ref 21,22) so the molecular aspects of the system are well-understood. Because of the methods used to precipitate the Gemini surfactant, radial outgrowth of the crystalline ribbons occurs from individual micelles that are densely aggregated into amorphous cores in the initial stages of precipitation.

The results section is essentially a conventional study of the morphology and growth of the asters (Figs 2 and 3) followed by mechanical properties (Fig 4) and micro-manipulation (Fig 5). This is nice work but I did not see any significant advances in methodologies or novel observations and insights.

The work on size selectivity of the auxiliary particles was interesting but again was limited by being descriptive and did not go beyond simply categorizing the observations.

Give the limited scope of the experimental work, the authors attempt to increase the relevance of the results by reference to microtubule asters and biomimetics. I was not convinced that this was appropriate and the conflation of microtubule dynamics (which involves oriented attachment of tubulin dimers, competitive assembly and disassembly at the +/- ends, force generation etc) with the static supramolecular assembly of a single component surfactant was too fanciful in my opinion. Similarly, comparing the position of the D particles to chromosomes in the mid region of spindles in cell division was unrealistic.

Thus I found the basic claims of the paper and the notion that the synthetic asters were analogous to radial skeletons to be contentious.

Finally, the development of complex higher order architectures from amorphous aggregates of micelles has been documented in inorganic/surfactant and inorganic/polymer systems studied many years ago by Coelfen, Antonietti and Mann, who worked on Ba[AOT]₂/SO₄²⁻-and Ca phosphate/polymer systems, Ozin who worked on silica/surfactant mesostructured complex forms, and more recently by Aisenberg (in a different but related context [BaCO₃/silica]). Specifically, the formation of calcium phosphate/polymer asters was reported in 1998 (Antonietti M, et al. Inorganic-organic Mesostructures with Complex Morphologies: Precipitation of Calcium Phosphate in the Presence of Double-hydrophilic Block Copolymers. Chem. Eur. J. 4, 2491-2498 (1998)). None of this much earlier work or the attendant reviews at the time on "morphosynthesis" were cited.

In conclusion, the manuscript would be much better placed in a more specialized journal focused on soft matter.

Synthetic asters as elastic, radial skeletons

Response to Reviewer # 1

We thank the reviewer for the insightful comments and the recommendation of publication after revision. We have done our best, in the revised manuscript, to address the reviewer's concerns.

1. Concerns about polarity and active transport of the current synthetic asters.

“Presumably, the synthetic asters also exhibit some sort of polarity, although this property is not explicitly characterized. However, in addition to the positioning of passive particles, it would be interesting to make a connection to the transport of active particles.”

We agree that polarity and active transport are two very important aspects of microtubule asters. For polarity, the building unit, tubulin is inherently asymmetrical such that the microtubules are of a plus and a minus end. For fiber-forming synthetic molecules (including ours in this paper), they are usually symmetrical in the fiber direction or the association/dissociation rate at two fiber ends are identical, so the fibers have no polarity.

For active transportation, motor proteins (like dynein and kinesin) bind to microtubules in very specific manners and move by consuming ATP. It would be fascinating to fabricate artificial motors that interact with our synthetic asters in similar ways. As a simple trial, we managed to grow asters in bacterial solutions to see if the bacteria can swim along the astral ribbons. Unfortunately, the bacteria (*E. coli* and *Staphylococcus aureus*) do not interact with the aster ribbons and are simply excluded from the asters. We plan to design and synthesize new active particles that can specifically interact with the astral ribbons as a follow-up project. These notes are added to the discussion section.

2. Concerns about the possibility of aster formation in the absence of centrosomes.

“there are also experimental demonstrations of asters formed spontaneously even in the absence of centrosomes, Ref. 6. By the way, the name of the first author of Ref 6 is misspelled, it should be Nedelec. I am wondering if there is a similar pathway for synthetic asters.”

Indeed, microtubule asters have been observed to form in the absence of centrosomes as a result of motor-microtubule interactions [Surrey, T. et. al. (2001). Physical properties determining self-organization of motors and microtubules. *Science*, 292(5519), 1167-1171, now cited in the revised manuscript]. However, the presence of cores at a proper concentration is indispensable for aster formation

in our case. For synthetic molecules, their assembly into fibers is usually dictated by the classic nucleation-growth mechanism that, in the absence of organizing centers, would produce a percolating network (gels) of randomly arranged fibers with few exceptions. In contrast to the classic mechanism, fiberization here is preceded by the relatively long-lived, yet metastable amorphous nodes (Figure 3c). They play a crucial role in aster formation by constituting the organizing centers (cores, Figure 3d) and eventually transforming into crystalline, radial ribbons. Although the core size varies, we do not observe any core-free asters. In addition, externally added hydrophobic particles can serve as heterogeneous nucleation sites facilitating core formation around them. Adding too many particles (particle/surfactant weight > 0.1) produces small, incomplete asters and favors fiber networks. Therefore, we think the cores are crucial for aster formation. This discussion is added to the section “Multi-stage mechanism of aster formation” in the revised manuscript.

Thanks for noticing the misspelling, we have fixed in the revision.

3. Concerns about the premature characterization on the mechanical properties of asters.

“The part on the mechanical properties of asters is rather incomplete. It is not clear what insights provides macroscopic rheology of aster precipitate phase since the elastic moduli depend on the aster concentration, positions, etc. The origin of hysteresis in Fig 4n is not clarified: I am not sure what does “a finite energy dissipation” mean in this context. Fig. 4o demonstrates a variation of the elastic modulus E as a function of position with respect to the center. Can a simple mechanical model explain this dependence?”

We thank the reviewer for the constructive comments and significantly expand the section on mechanical properties with new experimental results (see the figure below). For AFM measurements, there is certain hysteresis between indentation (black) and retraction (grey) curves (panel a, below). During indentation, some astral ribbons might be partially attached to the AFM tip by non-specific interactions. During retraction, they will gradually detach from the AFM tip. This attachment/detachment dissipates energy and is the main reason for hysteresis.

This figure is now Fig. 5 in the revised manuscript.

The modulus drops remarkably when the probing spot is moved from astral center to edge (panel b and c, below). We follow the reviewer's advice and put forward a simple model to describe this dependence. First consider a single astral ribbon as a semiflexible filament of contour length ℓ and persistence length ℓ_p (panel d, below). In the linear regime, the filament is characterized by two moduli transverse and parallel to the filament, respectively,

$$k_{\perp} = 3\kappa/\ell^3 \quad (1)$$

$$k_{\parallel} = 6\kappa^2/(k_B T \ell^4), \quad (2)$$

where κ is related to persistence length ℓ_p by

$$\ell_p = \kappa/(k_B T). \quad (3)$$

The filament is clamped at one end with a fixed position and a fixed orientation. A force acting on the other end can deform the filament. When the force is parallel and transverse to the filament, respectively, the deformation is given by

$$\Delta_{\parallel} = F_{\parallel}/k_{\parallel} \quad (4)$$

$$\Delta_{\perp} = F_{\perp}/k_{\perp} \quad (5)$$

When the filament is placed at an angle of θ , F is decomposed into $F_{\parallel} = F \sin \theta$ and $F_{\perp} = F \cos \theta$, and deformation in the force direction reads,

$$\Delta = \Delta_{\parallel} \sin \theta + \Delta_{\perp} \cos \theta = \frac{F \sin^2 \theta}{k_{\parallel}} + \frac{F \cos^2 \theta}{k_{\perp}} \quad (6)$$

Combination of eq. 1, 2, 3, and 5 leads to

$$\frac{F}{\Delta} = k_{\parallel} / [1 + (\frac{2\ell_p}{\ell} - 1) \cos^2 \theta] \quad (7)$$

In our case, the filament is quite rigid so $\ell_p \gg \ell$ and eq. 6 reduces to

$$\frac{F}{\Delta} = k_{\parallel} / (1 + \frac{2\ell_p}{\ell} \cos^2 \theta) \quad (8)$$

Next we consider the geometry of AFM measurements on a single aster (radius R) with the probing spot at a lateral distance d away from the center. The average orientation of ribbons in the probing spot is given by $\cos \theta = d/R$. And we assume the effective number density (A) of ribbons in contact with the AFM probe is constant for different d . The observed elastic modulus (E) measures the ratio of force to deformation,

$$E = \frac{F_{total}}{\Delta} = \frac{AF}{\Delta} = Ak_{\parallel} / (1 + \frac{2\ell_p}{\ell} \cos^2 \theta) \quad (9)$$

Given eq. 2 and $\ell = R$, we finally reach

$$E = 6k_B T A \ell_p^2 k_{\parallel} / [R^4 (1 + \frac{2\ell_p}{R} \cos^2 \theta)] \quad (10)$$

Now we compare eq. 10 (black curve in panel c) with the experimental results (color dots). In two limiting cases of $d \rightarrow 0$ and $d \rightarrow R$, the ratio of E reads,

$$\frac{E_{d \rightarrow 0}}{E_{d \rightarrow R}} = \frac{2\ell_p + R}{R} \quad (11)$$

For this particular aster, $R \approx 50 \mu\text{m}$ and the ratio is 40 from panel c, giving $\ell_p \approx 1 \text{ mm}$. Notably, this persistence length is quite close to that of microtubules (4 mm). With R and ℓ_p determined, the shape of the curve is fixed and A merely shifts it vertically (panel c). A simple fitting matches the experimental points remarkably well, giving $A = 1 \mu\text{m}^{-2}$, a reasonable value.

The reversible bending behavior up to $\pm 180^\circ$ as manipulated by an optical tweezer is demonstrated in panel e. The concentration dependence of aster gels is displayed in panel g.

Response to Reviewer # 2

We thank the reviewer for the critical comments on the novelty of this work and the analogy to biological asters. The manuscript is now substantially revised to emphasize the differences between the current asters and previously reported core-ray structures and to justify the analogy.

1. Concerns about the resemblance between the biological and synthetic asters.

“...I was not convinced that this was appropriate and the conflation of microtubule dynamics (which involves oriented attachment of tubulin dimers, competitive assembly and disassembly at the +/- ends, force generation etc) with the static supramolecular assembly of a single component surfactant was too fanciful in my opinion. Similarly, comparing the position of the D particles to chromosomes in the mid region of spindles in cell division was unrealistic.” “...the synthetic asters were analogous to radial skeletons to be contentious”

We acknowledge that microtubule asters are indeed complicated, dynamic structures that cannot be fully paralleled by the current synthetic asters. However, we propose the current asters as a rudimentary mimicry, especially in terms of elasticity and positioning capability, and support this proposition by experimental results.

- 1) For elasticity, it is critical for the microtubule asters to sustain a finite cellular or subcellular stress. The elasticity was well characterized for single microtubules or synthetic fibers, but not on single-aster level to the best of our knowledge. To this end, we have performed AFM measurements for single asters, revealing cell-like elasticity. In addition, new results in this revision shows that the persistence length of astral ribbons is on the order of 1 μm , quite similar to that of microtubules (Fig. 5d and relevant discussion in the main text), further supporting our claim.
- 2) For particle positioning, it was rarely reported in literature for artificial systems and was quite unique to the biological systems. Our results suggested that positioning can be achieved by a simple combination of particle size and surface chemistry without invoking any complicated biological machinery. New quantitative data is provided in Figure 4b in the revised manuscript.
- 3) Simplicity of the current system. As the reviewer suggested, the aster here is but a supramolecular assembly of a single component surfactant. Actually, simplicity of this system is of the very essence here if one intends to take it as a starting point to fully replicate biological asters by gradually incorporating other components and functionality into it.
- 4) There have been no reported attempts on manipulating multiple biological asters or previous core-ray particles. We maneuver the synthetic asters, beyond their biological counterparts, as building

blocks to form higher-order, tissue-like structures in virtue of aster-aster adhesion induced by ribbon intertwining.

- 5) As for the polarity, dynamic instability, and treadmilling of microtubules, they are not replicated in this work. We admit that the current asters are only incomplete, partial mimicry of the biological ones, but they can be improved with additional efforts. Taking protocell mimicry as an example, people started with simple lipid or fatty acid vesicles first, and added different components (RNA, polyelectrolytes, proteins, etc) and functionalities (division, replication, sensing, etc) to them in a step-by-step fashion. The entire process towards complete mimicry may take decades and many group's efforts. It would be unrealistic to perfectly replicate the biological asters in one work.

Overall, we hope that this work can draw people's attention in nanoscience to the biological asters and that this system can act as a prototype into which more functions can be incorporated. Part of this discussion is added to the revised manuscript.

2. Concerns about previously reported core-ray structures.

“documented in inorganic/surfactant and inorganic/polymer systems studied many years ago by Coelfen, Antonietti and Mann, who worked on Ba[AOT]₂/SO₄²⁻-and Ca phosphate/polymer systems, Ozin who worked on silica/surfactant mesostructured complex forms, and more recently by Aisenberg (in a different but related context [BaCO₃/silica]). Specifically, the formation of calcium phosphate/polymer asters was reported in 1998 (Antonietti M, et al. Inorganic-organic Mesostructures with Complex Morphologies: Precipitation of Calcium Phosphate in the Presence of Double-hydrophilic Block Copolymers. Chem. Eur. J. 4, 2491-2498 (1998)). None of this much earlier work or the attendant reviews at the time on “morphosynthesis” were cited.”

We apologize for not citing these highly relevant works, they are now cited and properly addressed in the revised manuscript. As mentioned by the reviewer, there have been a few reports since the early “morphosynthesis” time. For example, Sokolov et. al. fabricated mesoporous silica featuring radial patterns and Antonietti et. al. synthesized inorganic/organic hybrid particles of neuronlike morphology. More recently, the paradigm has been shifted to emerging functionality of the radial geometry. For instance, Kaplan et. al. demonstrated light-guiding properties of microsculptures on a substrate with a hemi-astral geometry. Wang et. al. discovered that TiO₂ spiky particles in contact with cells can physically activate innate immunity, echoing the effect of spiky nanostructures on virus or bacteria surfaces. This discussion is now added to the introduction section.

We select a few representative reported core-ray particles and compare them to the microtubule asters and the current synthetic asters in the following table.

	Ref.	Structure				Property		Function		
		Size (μm)	Ray den.	Heli.	Pola.	Elasticity	Dyn.	Pos.	Act.	
Inorganic	Microtubule asters	2, 4	5 to 100	moderate	Y	Y	semiflexible	Y	Y	Y
	Synthetic asters (this work)		50 to 120	moderate	Y	N	semiflexible	N	Y	N
	BaCO ₃ /SiO ₂ hemi-asters	10	~50	moderate	Y	N	rigid	N	NA	N
	ZnO Hedgehog particles	15	~1	moderate	N	N	rigid	N	NA	N
	SiO ₂ spiky particles	11	~1	moderate	N	N	rigid	N	NA	N
	Organic	Protein spherulites	14	20 to 100	compact	Y	N	rigid	N	N
Malonamide spherulites		13	~100	compact	N	N	rigid	N	N	N
Coacervate-based asters		18	~50	moderate	Y	N	soft	N	NA	N
Hybrid		Ca ₃ (PO ₄) ₂ -polymer particles	9	0.2	low	N	N	NA	N	NA

This table is now Supplementary Table 1. Abbreviations: Ref. = reference; Ray den. = ray density; Heli. = helicity; Pola. = polarity; Dyn. = dynamic nature; Pos. = positioning capability; Act. = active transportation; Y = yes; N = no; NA = not available.

The biological asters feature helical, polar, semiflexible microtubules radiating from a center at a moderate density. The microtubules are semiflexible filaments with a persistence length ~ 4 μm , which can sustain a certain stress and deform to an extent. The microtubules are dynamic filaments that constantly undergo assembly/disassembly by consuming ATP. The asters are crucial in subcellular positioning and active transportation.

In comparison, the synthetic ones can be roughly categorized into inorganic, organic, and hybrid particles. They are markedly different in size, ray density, elasticity, and functionality. Many synthetic ones are made of rigid rays that cannot deform under a biologically relevant force; the coacervate-based asters are too soft to sustain certain stress. Two spherulites are too compact to accommodate any particles precluding their application in particle positioning. The BaCO₃/SiO₂ structures were fabricated by “microsculpturing” on a substrate, so they are attached to the substrate and are only hemi-asters. The Ca₃(PO₄)₂-polymer hybrid particles are 200 nm in size too small to position other particles. Overall, the main advance in this work is that the current asters feature cell-like elasticity and capability in particle positioning, whereas the previous core-ray structures fall short in these two aspects. Notably, all the synthetic asters cannot parallel the biological ones in terms of polarity, dynamic nature, and active transportation yet. These notes are added to the discussion section in the revised manuscript.

3. Concerns about the descriptive results on particle positioning.

“The work on size selectivity of the auxiliary particles was interesting but again was limited by being descriptive and did not go beyond simply categorizing the observations.”

To quantitatively resolve particle distribution, we employ a particle tracking program (TrackMate on ImageJ) to obtain particle coordinates from the fluorescence channel and identify aster center and radius R from the transmission channel (the following figure, left panel). Lateral distance of a particle from the center d is determined and statistics of d/R for different types of particles is established (the following figure, right panel). Clearly, from type A to D, the particles are sharply localized in the core, widely distributed along the ribbons, preferentially placed near the outer rims, and constrained to the periphery. This positioning scheme thus offers a facile means to compartmentalize mixed components, which are difficult to separate otherwise, in an efficient and orthogonal manner. This discussion is now added to the main text.

The left panel is now Supplementary Figure 4 and the right panel now Figure 4b.

Reviewers' comments:

Reviewer #1 (Remarks to the Author):

Authors addressed all my concerns, I have no further objections against acceptance of this work

Reviewer #2 (Remarks to the Author):

Although the authors have included further results and references that help to strengthen the paper, I am still of the opinion that the authors undermine the high quality of their experimental work by their resolute claim that the morphologically complex surfactant precipitates have biomimetic relevance to microtubules ("Herein, we report a class of synthetic asters that mimic microtubule asters in morphology, mechanics, and positioning capability.").

To identify their work as "mimetic", the authors have to demonstrate the representation of at least one essential feature of microtubules in the synthetic materials. Comparing gross morphologies and generic mechanical properties do not address this requirement. Morphological similarity does not necessarily imply the presence of structural, functional, organizational or mechanistic interrelationships between the materials under consideration. The same applies for mechanical properties; strong statements such as, "Therefore, the current astral system parallels biological systems in mechanical properties on three hierarchical levels; the ribbons and microtubules are of similar persistence lengths, the synthetic asters and cells of comparable young's moduli, and multi-astral gels and soft tissues of similar shear moduli." remain highly contentious in my opinion. Similarly, the "mimicking" of "positioning capability" is spurious given that the key aspect of microtubule activity is the dynamics of their non-equilibrium state, rather than the passive size sorting (chemical/physical adhesion/expulsion) observed for the synthetic system.

I therefore remain of the opinion that there is virtually nothing of relevance that connects the biological and synthetic materials under discussion. I appreciate the authors attempt to address some of these points in the Discussion and indeed temper their claims to a rudimentary form of mimicry; but why set up the grandiose claims in the first place if they are not well substantiated?

An alternative approach would be to remove all the text in the Introduction and Results that pertain to the microtubule-based motivation, and focus the objectives on the preparation and properties of synthetic supramolecular materials with complex aster-like morphologies. In fact, the title "Synthetic asters as elastic, radial skeletons" fits this scenario very well. Given this approach, it would then be reasonable to include some of the putative comparisons with microtubules currently described in the Introduction and Results as part of an extended Discussion section. I appreciate this requires quite a lot of work for the authors but I do believe this would give the paper more credibility and standing in the scientific community. In the absence of such major revisions, and given the relatively low conceptual advance of the work, I cannot support publication of this manuscript in NCOMM.

Revision Requested for Manuscript NCOMMS-19-18860A

Synthetic asters as elastic, radial skeletons

Response to Reviewer # 1

We are glad that the reviewer is satisfied with the revision.

Response to Reviewer # 2

1. The reviewer approved the quality of our experimental work and suggested us to drop the claims on biomimicry.

“I am still of the opinion that the authors undermine the high quality of their experimental work by their resolute claim that the morphologically complex surfactant precipitates have biomimetic relevance to microtubules.”

“An alternative approach would be to remove all the text in the Introduction and Results that pertain to the microtubule-based motivation, and focus the objectives on the preparation and properties of synthetic supramolecular materials with complex aster-like morphologies. In fact, the title “Synthetic asters as elastic, radial skeletons” fits this scenario very well. Given this approach, it would then be reasonable to include some of the putative comparisons with microtubules currently described in the Introduction and Results as part of an extended Discussion section.”

We thank the reviewer for the insightful suggestion and agree that the claims on mimicry of microtubule asters are premature. As the reviewer suggested, we have now changed the narrative in this revision. Specifically, we dropped all the claims on biomimicry throughout the paper. Texts pertaining microtubule asters were removed from the Abstract, Introduction, and Results sections, except for one sentence in the first paragraph in the Introduction where we state it as an example of the core-ray structure in nature. We do not make any connection between the synthetic and biological asters in these 3 sections.

The previous texts are now moved to the Discussion section, where we described microtubule asters first, and then laid out the similarities and differences of the synthetic and biological asters in morphology, formation mechanism, mechanics, and positioning capability. The fact that the synthetic asters are static in nature is stressed. We avoided the use of “mimicry”, “mimetic”, or “parallel”, and simply state objective results. References and relevant texts were rearranged to make this narrative smooth. We hope that the revision can bear more credibility and meet the reviewer’s criteria.